# Prediction of Reservoir Fracture Parameters Based on the Multi-Layer Perceptron Machine-Learning Method: A Case Study of Ordovician and Cambrian Carbonate Rocks in Nanpu Sag, Bohai Bay Basin, China

**Jianya Pei [1,2,\*] and Yunfeng Zhang [1]**

1 The School of Earth Science, Northeast Petroleum University, Daqing 163453, China
2 Logging & Testing Service Company, Daqing Oilfield Company Limited, Daqing 163453, China
\* Correspondence: peijy0421@126.com; Tel.: +86-459-5576-135

**Abstract:** Developing a model that can accurately predict internal fractured reservoirs in the context of the ultra-low physical properties of carbonate rocks by only employing conventional mathematical methods can be very challenging. This process is challenging because the relationship between basic fracture parameters and the logging response in carbonate reservoirs has not been studied, and the traditional method lacks adaptability due to the complex relationship between basic fracture parameters and the logging response. However, data-driven approaches supplemented by machine learning algorithms based on multi-layer perceptrons (MLP) provide a more reliable solution to this challenge. In this paper, a classical fracture parameter evaluation data set is established using fracture porosity, fracture density, fracture length, and fracture width data that can be identified by resistivity and acoustic imaging logging. Another data set can be composed of different types of logs, and it can be used to identify reservoirs. Two different data sets were validated by regression task evaluation indicators in machine learning, and the correlation coefficient $R^2$ is greater than 0.82. This means that the model accuracy of the algorithm can reach 82%. Combined with the comparison results of eight conventional machine learning algorithms, the reliability and application validity of the MLP model are verified. This method's accuracy is also verified by oil test data, which show that the MLP machine-learning algorithm can effectively simulate the relationship between lithology and fracture development. In addition, it can be used to predict key exploration horizons before drilling. The relationship between lithology and fracture development degree is well-simulated by the MLP machine learning algorithm, which shows that the degree of fracture development is mainly affected by fractures, indicating that the method can be used to predict key exploration horizons before drilling.

**Keywords:** fracture parameter prediction; MLP machine learning method; Ordovician carbonate; Cambrian carbonate



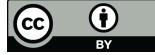

## 1. Introduction

Fractured reservoirs are an important research field in terms of increasing production and reserves in the 21st century, and fractured low permeability reservoirs are particularly important. The production from this type of reservoir accounts for more than half of the total oil and gas production [1]. For carbonate reservoirs with a matrix porosity of less than 5%, fractures act as percolation channels for fluid migration, but they are also favourable for low-permeability carbonate reservoirs. However, they are difficult to find because of deep burial, complex tectonic stress, differential diagenesis, and strong heterogeneity [2]. Previous research and exploration into fracture parameter characterization, identification, and prediction have been conducted [2]. Fracture parameter characterization is a key indicator for evaluating fracture development, including fracture occurrence, density, penetration depth, opening, length, filling characteristics, fracture porosity, and

permeability [2]. Because of the high cost of coring, the application of seismic and logging data in fracture identification and prediction has always been a hot research topic in this field. At present, there are two primary fracture prediction methods based on logging curves: (1) Commonly used logging methods that identify fractures include lithology logging, porosity logging, resistivity logging, acoustic full-wave logging, formation dip logging, and imaging logging [3–6], and (2) machine learning methods including the grey theory prediction method [7] and the neural network prediction method [8,9]. In short, based on conventional well log data, a total of approximately 50 kinds of fracture identification methods have been proposed, including the pore structure index [10], permeability difference [11], the resistivity correction difference ratio method [12], the resistivity instruction method [13], the ultrasonic shear wave splitting method [14], the comprehensive probability density method [15], the well-logging curve rate method [16], the logging curve reconstruction method [17], the maximum entropy prediction error principle method [18], the R/S analysis method [19], the fracture probability model method [20], the wavelet multiscale transformation method [21], the grey theory for fracture prediction [7], neural networks for fracture prediction [22,23], and the fractal dimension method [24].

Previous research, however, concentrated more on the carbonate reservoir's fracture porosity, while cracks in the basic parameters and the well-logging response have not been studied thoroughly in terms of locating carbonate reservoirs [2]. In addition, it is difficult to determine the relationship between basic fracture parameters and conventional logging responses due to the complex pore structure and severe heterogeneity (such as the influence of bitumen quality on reservoir physical properties [25]) of carbonate reservoirs. However, machine learning has great adaptability in terms of solving such nonlinear problems and can realize the transformation of complex functional relationships. In addition, relatively good results have been achieved in carbonate reservoir filling identification [26], lithology identification [27], complex reservoir fluid identification [28], horizon interface identification [29], and other aspects. Therefore, machine learning provides us with the possibility of solving the relationship between fracture parameters and the logging response.

The multi-layer perceptron (MLP, the relevant abbreviations are summarized in Schedule 1) algorithm was developed based on the perceptron model proposed by McCulloch and Pitts, and it is a supervised machine learning method. Its feedforward structure consists of one input layer, multiple hidden layers, and one output layer. The hidden layer is composed of one or more nonlinear neurons, and it has the ability to approximate any nonlinear relationship between the input layer and the output layer with arbitrary precision as well as high fault tolerance and robustness. In addition, it can comprehensively apply continuous variables and discrete variables to solve complex nonlinear problems [30].

The Cambrian and Ordovician systems in Nanpu Sag of Bohai Bay Basin have emerged as key exploration and development blocks in recent years, and they are one of the most important exploration targets in Bohai Bay Basin as well as the primary target area in this study. The exploration of weathering crust reservoirs in this area has shown great commercial value [31–34]. At present, oil and gas shows have also been found below the weathering crust reservoir, indicating that the internal-type reservoir also has good oil and gas development value. Due to the lack of exploration, well coring, and imaging logging data, research on the characteristics of internal-type reservoirs is still shallow, and to a large extent, this lack has hindered oil and gas exploration in the area.

In this paper, the fracture parameters of inner reservoirs are comprehensively studied by using the limited data related to resistivity and acoustic imaging logging evaluation. At the same time, the MLP machine-learning algorithm is used to model the reservoir fracture parameters in combination with conventional logging data. Then, the technical advantages of the machine learning method are discussed by comparing it with traditional methods. Finally, the location of a high-quality reservoir is predicted according to the mineral composition of each layer obtained from the outcrop, which provides a scientific basis for fine exploration and development of carbonate internal reservoirs.

## 2. Geological Background

### 2.1. Tectonic Characteristics

The Nanpu Depression is a dustpan-shaped depression that developed at the base of the North China platform, which experienced two periods of tectonic movement. Among them, the Nanpu 1, 2 and 3 buried hills are located on the slope belt of Nanpu Sag, and the Shaletian uplift, the Nanpu 4 buried hill is located on the hanging wall of the Baigezhuang fault, which is the boundary fault of Nanpu Sag, and the Nanpu 5 buried hill is located on the hanging wall of the southwest Zhuang Arc fault, which is the Nanpu Sag boundary fault (Figure 1). The buried hills in the Nanpu Depression are affected by multi-stage tectonic movement. Since the deposition of the Middle Neoproterozoic, the Indosinian, Yanshan, and Himalayan movements have produced multi-stage unconformity. The basic morphology of Nanpu's buried hills formed primarily during the Yanshan Period, while the fault activity in the controlling mountain began during the Himalayan period. This activity controlled the basic morphology of buried hills' structures. The Nanpu 1, 2, 3 and 5 buried hills are the primary exploration targets in the study area, and the pre-Cambrian–Ordovician stratigraphic system is the primary target. This system can be divided into two types of reservoirs: the weathering crust type and the internal type. "Weathering crust reservoir" refers to the oil and gas pools that gather at the top of a buried hill, and after years of exploration, most of the weathering crust-type oil and gas reservoirs have been gradually identified. An internal reservoir is one that is located approximately below the top surface of the weathering crust. Compared with weathering crust reservoirs, internal reservoirs are more heterogeneous and difficult to discover due to their deep burial, complex tectonic stress, and differential diagenesis. So far, the development degree of structural fractures is considered to be one of the primary controlling factors for the development of internal reservoirs.

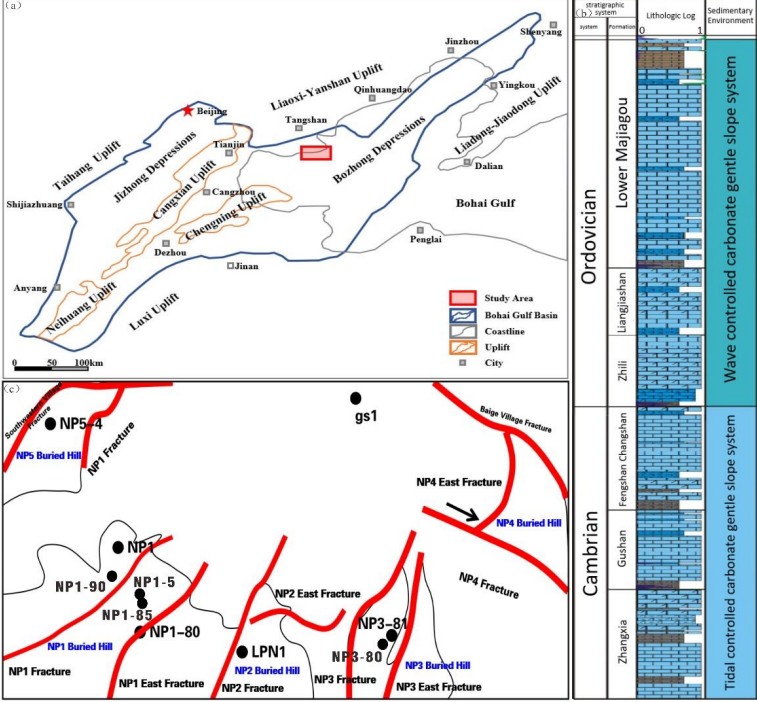

**Figure 1.** Comprehensive map of the regional geology. (**a**) Tectonic division; (**b**) The lithology characteristics; (**c**) Well location in study area.

### 2.2. Stratigraphic Characteristics

The buried hill strata in the Nanpu Depression have rapid lateral variation. From east to west, the stratum changes from old to new. The Nanpu 1 and 2 structures primarily developed the Ordovician–Cambrian strata (Figure 1b,c), while the Nanpu 3 structures

primarily developed the Cambrian strata. The lower region of the Cambrian Zhangxia Formation consists of thick, massive oolitic limestone; the middle region is interbedded thin limestone and yellow-green shale, and the upper region is interbedded leopard-spotted limestone and shale. The lower region of the Gushan Formation is interbedded with yellowish green shale and light grey thinly bedded limestone, and the upper region is lobed bamboo limestone and argillaceous striped limestone interbedded with shale. The Changshan Formation is characterized by bamboo leaf limestone with circular oxidation patterns and argillaceous striped limestone. The Fengshan Formation is characterized by oolitic limestone, bamboo leaf limestone, and argillaceous striped limestone, with brownish-grey dolomite and algal dolomite at the top.

Other formations include the Ordovician Yeli Formation, which is primarily composed of brown-grey medium-thick bamboo leaf dolomite, micrite dolomite, and powderfine dolomite, and the Liangjiashan Formation, which is composed of brownish-grey thick bedded chert-bearing limestone with nodules and greyish-white and light-yellow marlite dolomite. The Majiagou Formation consists of two distinct sedimentary cycles from dolomite to limestone: (1) The dolomite is primarily composed of thin to medium-thick layers of micritic dolomite, micritic dolomite, and argillaceous dolomite mixed with brecciated dolomite; and (2) the limestone primarily consists of medium-thick microcrystalline limestone, micrite limestone, medium-thick cloud-spotted limestone, and chert-banded limestone (nodular).

### 2.3. Fractures

Fractures are an important reservoir space and provide migration channels in inner reservoirs. When the fractures are netted, and the internal channels are not filled by cemented materials, high-quality reservoirs can form. In the study area, the inner reservoir space fractures were affected by dolomitization, recrystallization, and dissolution during the early stage, and the differential diagenesis during the later stage increased the reservoir space and improved the reservoir connectivity. In addition, most of the study area includes tectonic fractures, which were nearly filled by calcite during an early stage of formation, and therefore, they have poor oil and gas migration and storage capacity. However, after being cut by hydrothermal calcite during a later stage, new oil and gas storage space and a migration channel formed (Figure 2).

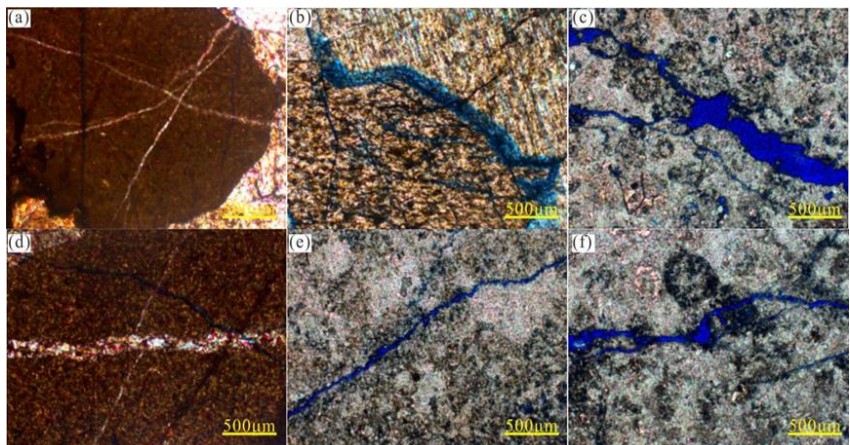

**Figure 2.** Fracture characteristics of Ordovician and Cambrian carbonate rocks. (**a**) Micritic limestone (well NP1-90, 5224.80 m); (**b**) silty limestone (well NP1-90, 5224.80 m); (**c**) dolomite (well NP3-80, 5682.59 m); (**d**) dolomite (well NP3-80, 5224.80 m); (**e**) micritic limestone (well NP1-90, 5682.59 m); and (**f**) dolomite (Well NP3-80, 5682.59 m). The advantage of cast thin sections over conventional thin sections is that the pore space is filled with stained resin or liquid glue, and the pore space can be directly and conveniently observed by a polarizing microscope while avoiding artificially induced pores and cracks.

### 3. Data Collection

In fractured reservoirs, the extension of a single fracture is insignificant, while the fracture development system formed by the combination of multiple fractures plays an important role in the formation of seepage channels. Therefore, it is necessary to evaluate the influence of fracture parameters and the fracture development system on reservoir seepage characteristics during reservoir fracture evaluation.

A comprehensive and systematic observation of fractures primarily includes the following parameters (Table 1): fracture density (FD), fracture length (FL), fracture width (FW), fracture porosity (FPOR), fracture orientation (FO), the fracture dip angle (FDA), and fracture filling (FF). Fullbore Formation Microresistivity Scanning Imaging Logging (FMI) from three wells were selected to determine the fracture parameters. In addition, because the orientation of fractures or fracture system development must be obtained by directional coring, conventional logging cannot determine fracture type or fracture filling. Therefore, this paper focuses on four important parameters: fracture density, length, width, and fracture porosity.

**Table 1.** Resistivity imaging logging fracture parameter range.

| Reservoir Fracture Parameters | Minimum Value | Maximum Value | Mean Value |
|---|---|---|---|
| Fracture porosity (%) | 0.001 | 0.47 | 4.69 |
| Fracture length (m/m$^2$) | 0 | 7.06 | 28 |
| Fracture width (μm) | 2 | 29 | 258 |
| Fracture density (1/m) | 0 | 6.15 | 28 |

In machine learning, the validity and representativeness of learning samples are two important factors for determining the prediction effect. Reservoir fractures are controlled by lithology, and various logging curves can reflect the physical characteristics of underground rocks from different angles. Therefore, the logging parameters that are sensitive to fracture parameters can be selected as learning samples for the prediction model based on the response characteristics of different logging data. The logging practice shows that the fracture characteristics of carbonate reservoirs show low RLLD, low gamma rays (GRs), low neutron gamma logging values, and high acoustic time differences. Therefore, GRs, acoustic curves (ACs), neutron porosity (CNL), neutron density (DEN), and deep lateral resistivity (RILD) were selected to construct the learning sample data set used in the prediction model.

For three selected wells (NP1-5, NP1-80, and NP1-85), four parameters were selected and calculated using FMI imaging logs as samples. The mean values of the GR, AC, CNL, DEN, and RILD logging data from the FMI imaging fracture sections were used as the petrophysical and electrical characteristics of the fracture samples for machine learning, and a total of 99 data points were recorded. The value range of the data points is shown in Figure 3. Due to the small number of data points obtained and the imbalanced distribution, the sample data was balanced, and the number of samples after the balancing process reached 350. Finally, the fracture parameter model was established by using the MLP algorithm after proper correction and normalization. The data point distribution before and after balancing is shown in Figure 3

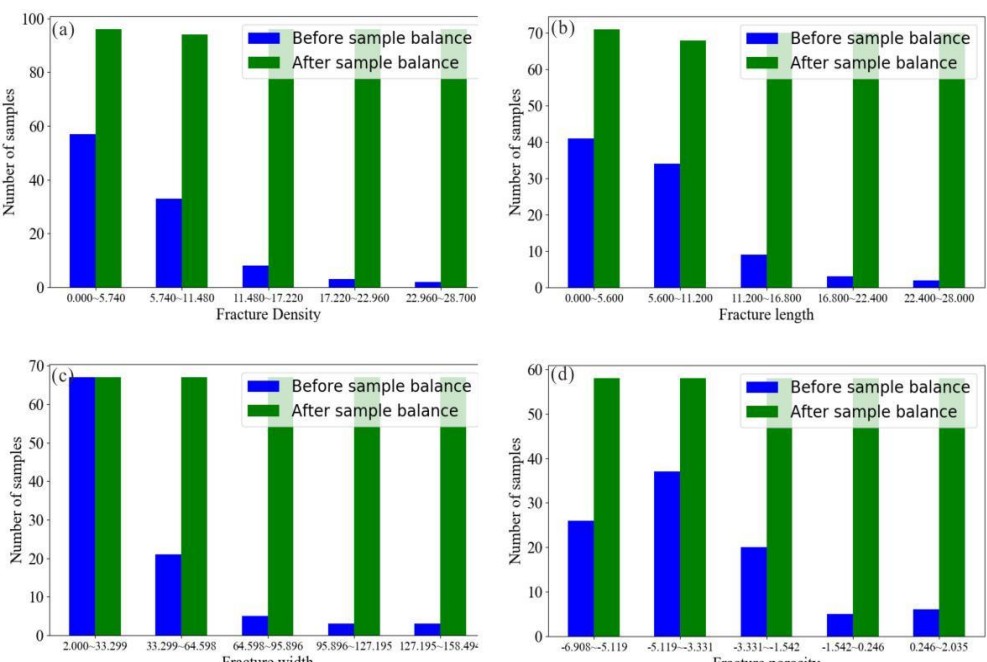

**Figure 3.** Modelling samples and data points after sample balance. (**a**) Fracture density; (**b**) Fracture length; (**c**) Fracture width; (**d**) Fracture porosity.

## 4. Methods and Workflow

### 4.1. SMOTE Algorithm

The basic idea of solving unbalanced data sets is to eliminate or reduce the data imbalance by changing the training data's distribution. The synthetic minority over-sampling technique is an algorithm that starts with minority samples, finds adjacent samples, synthesizes new minority samples, and maintains the number of minority samples so that they are consistent with the number of majority samples. However, the random linear interpolation method is adopted, which is blind, and the newly generated samples may not be accurate in terms of their appropriate position. The improved algorithm is based on a limited interpolation range, feature-weighted interpolation, and clustering interpolation, among other factors.

The sample collection process based on the SMOTE algorithm was as follows:

(1) The samples were divided into five intervals according to the size of the training objectives. The interval with the largest number of samples was selected as the majority category and recorded as a, while the samples in the other intervals were recorded as belonging to the minority category. The minority samples were recorded as b, c, d, and e. Class a was combined with classes b, c, d, and e in pairs.

(2) In each combination, for each sample in a minority class, the Euclidean distance was used as the standard for calculating its distance to all of the samples in the sample set of a minority class, and the minority class's k nearest neighbour was obtained. Then, several samples $x$ were randomly selected from the minority class's k-nearest neighbours. For each randomly selected neighbour xi, a new sample y was constructed with the original sample x according to the following formula:

$$y = x + \text{rand}(0.1) \times |x - xi| \tag{1}$$

(3) The new samples from all of the combinations were constructed and merged. Finally, the duplicate samples were deleted to obtain the new sample data set.

The SMOTE algorithm in the imBLearn library (https://github.com/topics/imblearn, accessed on 1 July 2022) was used to achieve sample balance.

### 4.2. Multi-Layer Perceptron Algorithm

Among the many machine learning architectures, the MLP neural network has been widely used for its simple structure, easy implementation, good fault tolerance, robustness, and strong nonlinear mapping ability. The data is received by the input layer through one or more fully connected layers, and the neurons in each fully connected layer can fit the original data. Finally, the data is output by the output layer. Then, the output value and the sample label are used to construct the loss function, and the error value of the loss function is iteratively reduced. Additionally, the model parameters are updated by the back-propagation gradient descent algorithm so that the loss function can reach an optimal value. At this point, the MLP model has the ability to accurately fit the sample features.

The specific flow of the MLP algorithm is as follows:

(1) During the forward propagation stage, assuming that there are m neural units in layer k-1 and n neural units in layer k, the weight matrix of layer k is expressed as $W^k \in R^{n \times m}$, and the bias matrix is expressed as $b^k \in R^{n \times 1}$. The initialization and calculation process can be divided into the following three steps:

Step 1: Initialization $\to$ o1 = x, k = 1;

Step 2: Order $\to$ k = k + 1, to calculate $\to o^k = \sigma(z^k) = \sigma(W^k o^{k-1} + b^k)$;

Step 3: Repetition $\to$ Step 2, up to k = K, output $o^K$.

(2) The back-propagation algorithm updates the weight, and in essence, it adjusts the weight along the direction of the mean squared error (MSE) decrease. The m training sample data sets are expressed as follows:

$$D = \{(x_1, y_1), (x_2, y_2), \cdots (x_m, y_m)\} \tag{2}$$

The total number of MLP layers is K, the gradient descent method's iteration step is $\alpha$, the maximum iteration number is MaxIter, and the threshold for stopping iteration is ⊚.

Step 1: Initialize the weight matrix W and bias matrix b for each hidden layer and output layer.

Step 2: For Iter, from 1 to MaxIter:

Step 2.1: For i, from 1 to m:

(1): Initialize the input o1 to $x_i$;

(2): For k = 2 to k, calculate the forward propagation:

$$o^{i,k} = \sigma\left(z^{i,k}\right) = \sigma\left(W^k o^{i,k-1} + b^k\right) \tag{3}$$

(3): The output layer is calculated by the loss function θi,k:

(4): For k = K − 1 to 2, the backpropagation calculation is performed:

$$\theta^{i,k} = \left(W^{k-1}\right)^T \theta^{i,k-1} \times \sigma'\left(z^{i,k}\right) \tag{4}$$

Step 2.2: If all of the changes in W and b are less than $\varepsilon$, then the program exits the cycle.

Step 3: Output weight matrix W and bias matrix b for each hidden layer and output layer.

### 4.3. Workflow

As shown in Figure 4, the prediction model's workflow for the crack parameters, which is based on SMOTE and the MLP algorithm, is divided into the following steps:

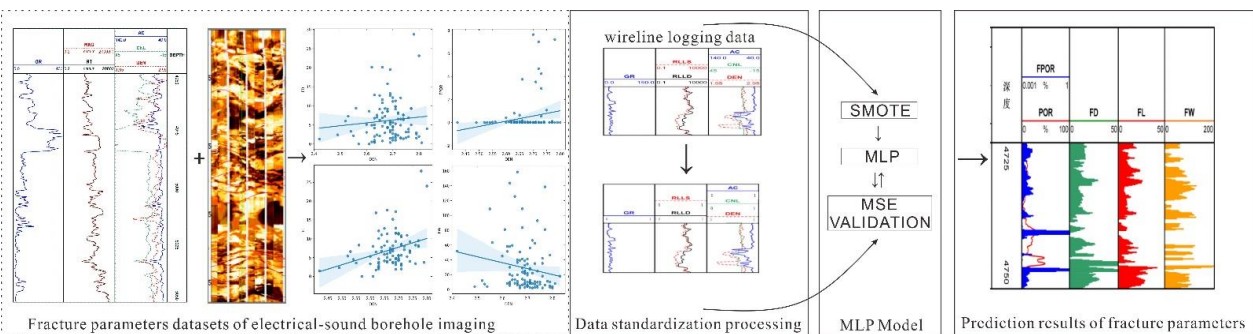

**Figure 4.** SMOTE and MLP-based reservoir fracture parameter evaluation model workflow. The dotted boxes denote the data collection process, and the solid frames represent fracture parameter prediction based on the MLP model. In this study, Python and the MLP algorithms from the scikit-learn library (http://scikit-learn.org, accessed on 1 July 2022) were used to predict the reservoir fracture parameters.

(1) The density, width, length, and porosity of reservoir fractures are detected and calculated using imaging logs and correlated with the conventional log to obtain a sample data set, where the conventional log includes the FD, FL, FW, and FPOR. This process is depicted on the left side of Figure 4.

(2) The SMOTE algorithm was used for sample balance to obtain a reasonable data set.

(3) The standard deviation (Z-Score) was used to standardize the dataset, including GR (API), AC (us/m), CNL) (%), DEN (g/cm$^3$), and RILD ($\Omega \bullet$m) logging. This process is depicted in the centre of Figure 4.

(4) During MLP model construction, a mesh search was used to optimize the hyper-parameters, and cross-validation was also conducted.

(5) The best MLP model for reservoir fracture parameter evaluation in the selected area was determined and is shown on the right side of Figure 4.

*4.4. Machine-Learning Capability Assessment*

In regression tasks, there is a large difference between the performance evaluation indexes of regression models and classification models. $Y_{true}$ represents the true value, and $Y_{pred}$ represents the predicted value. The four commonly used evaluation indexes primarily include the explained variance score (EVS), the mean absolute error (MAE), the MSE, and the R2 score.

(1) EVS: This metric is used to calculate the variance score of the regression model, and the value range is [0, 1]. The closer it is to 1, the better the independent variable can explain the variance in the dependent variable:

$$\text{explained\_variance\_score} = 1 - \frac{var\left(Y_{true} - Y_{pred}\right)}{var(Y_{true})} \tag{5}$$

MAE: The MAE is used to evaluate the degree of closeness between the predicted results and the real data set, and the smaller the value, the better the fitting effect:

$$\text{mean\_absolute\_error} = 1/n \sum |Y\_true - Y\_pred| \tag{6}$$

MSE: This metric is calculated as the mean of the sum of the squares of the errors for the sample points that correspond to the fitted data and the original data. The smaller the value, the better the fitting effect:

$$\text{mean\_squared\_error} = 1/n \sum \llbracket (Y\_true - Y\_pred) \rrbracket \verb|^|2 \tag{7}$$

R2 score: The R2 score can be understood as the proportion of the variation performance in dependent variable Y that can be explained by the estimated multiple regression equation, which measures the degree to which each independent variable can explain the variation of the dependent variable. Its value is between 0 and 1, and the closer its value is to 1, the better the variable can be explained:

$$r2\_score = 1 - \left( \sum [\![ (Y\_true - Y\_pred) ]\!] \, ^2 \right)/(n \times var(Y\_true)) \tag{8}$$

## 5. Results

### 5.1. Machine-Learning Capability Assessment

The establishment of the MLP model is divided into the following two steps:

Step 1: The sample set is divided into a training set and a testing set. The train_test_split module was used to select a subset of the features. The training sample size accounted for 80% of the total sample size, and the remaining 20% of the samples were used as test samples.

Step 2: In neural networks, the selection of hidden layers and neurons directly affects the model's performance. If the number of hidden layers and neurons is too small, the network will not have the necessary learning or information-processing abilities. On the contrary, if the network structure is too complex, the network learning rate will be low; additionally, over-fitting may occur, which affects the network's generalization. In addition, there is no clear theory or method regarding the selection of the number of hidden layers and neurons. In general, the number of hidden layer neurons in an MLP neural network can be determined by the following empirical formula:

$$N = \frac{N_{training}}{a(N_0 + N_1)} \tag{9}$$

where $N$ is the number of neurons in the hidden layer, $N_0$ is the number of neurons in the input layer, $N_1$ is the number of neurons in the output layer, and a is an integer ranging from 1 to 10. Ntraining is the number of training samples.

Based on equation (9) and the input data's dimension in the network's data set, a network architecture with two hidden layers and 10 neurons was finally selected to establish the model. The other parameters were set as follows: the learning rate was 0.01, the optimizer was the Quasi-Newton method (L-BFGS), and the number of training epochs was 2000.

### 5.2. Results of Model Training

The training results of the MLP model proposed in this paper are shown in Table 2. The explained_variance_score of the training set and the test set for the four fracture parameters are greater than 0.98 and 0.83, respectively. In addition, the goodness of fit scores were greater than 0.97 and 0.82.

**Table 2.** Training results of the MLP model.

| Fracture Parameters | Evaluation Index of the MLD Model Test Data Set | | | | Evaluation Index of the MLD Model Training Data Set | | | |
|---|---|---|---|---|---|---|---|---|
| | EVS | MAE | MES | R2_score | EVS | MAE | MSE | R2_score |
| FD | 0.856 | 0.197 | 0.127 | 0.856 | 0.980 | 0.091 | 0.020 | 0.980 |
| FL | 0.834 | 0.259 | 0.212 | 0.821 | 0.982 | 0.078 | 0.018 | 0.982 |
| FW | 0.837 | 0.270 | 0.212 | 0.826 | 0.984 | 0.081 | 0.016 | 0.984 |
| FPOR | 0.886 | 0.185 | 0.102 | 0.886 | 0.962 | 0.119 | 0.038 | 0.962 |

To investigate the accuracy of the model's prediction in different numerical ranges, the fracture parameters were divided into two ranges and compared. The comparison between the measured values and the predicted values for the test set partitions of the MLP model shows that (Figure 5) the measured fracture densities were divided by 15 fractures

per meter, and the goodness of fit was 0.76 and 0.969 for the training set and the test set, respectively. The fracture length was measured with 15 m per square meter as the dividing line, and the goodness of fit was 0.804 and 0.963 for the training set and test set, respectively. These results show that the prediction accuracy for high fracture densities and lengths is relatively high. However, the measured fracture width was separated by 75 μm, and the goodness of fit was 0.94 for the training set and 0.792 for the test set; the fracture porosity is bounded by 0.4%, and the goodness of fit is 0.972 and 0.974 for the training set and the test set, respectively. Therefore, the prediction accuracy for large fracture widths is relatively low.

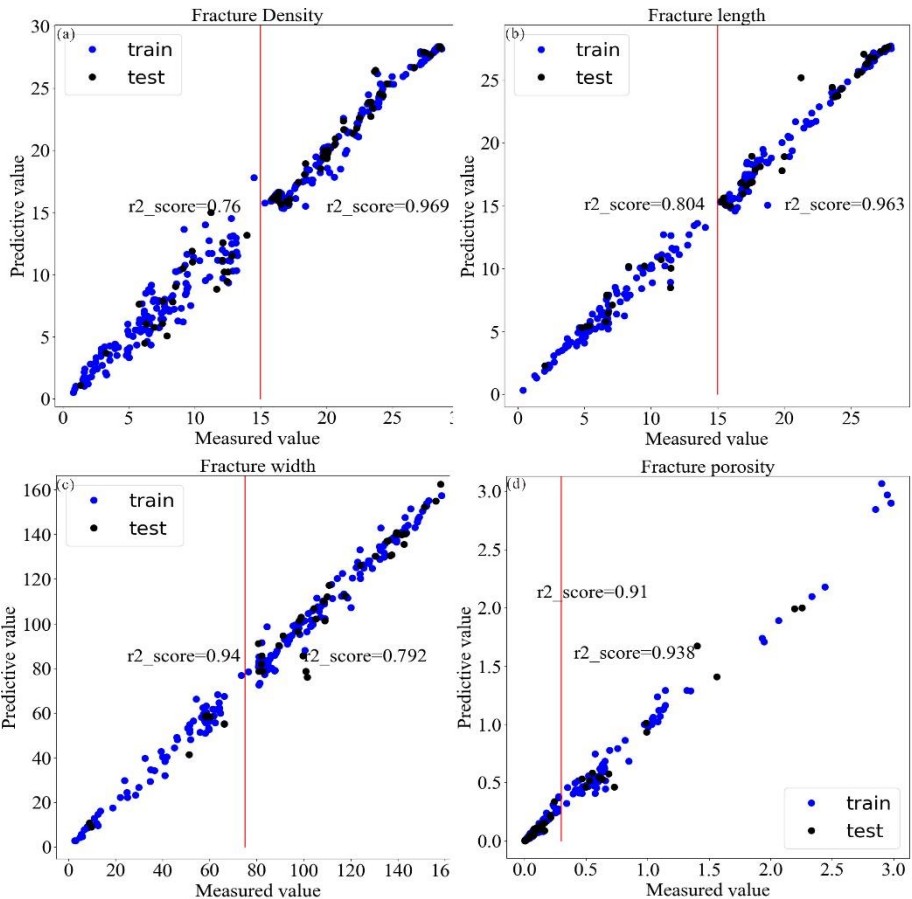

**Figure 5.** Comparison between the measured and predicted fracture parameters. (**a**) Fracture density; (**b**) Fracture length; (**c**) Fracture width; (**d**) Fracture porosity.

In addition, Figure 6 shows the comparison between the fracture development degree shown by the imaging logging and conventional logging predictions for the Laopu South 1 well (not modelled). The density, length, width, and porosity of the fractures were predicted to be higher for the first-order fracture segments. The predicted density, length, width, and porosity of the fractures are not significantly different between the secondary and tertiary fracture segments. These results are consistent with the comparison results between the measured and predicted fracture parameters. In other words, the multi-layer perceptron algorithm applied in this paper is very suitable for fracture parameter evaluation of carbonate reservoirs, and the model has high accuracy and high reliability in terms of its prediction results.

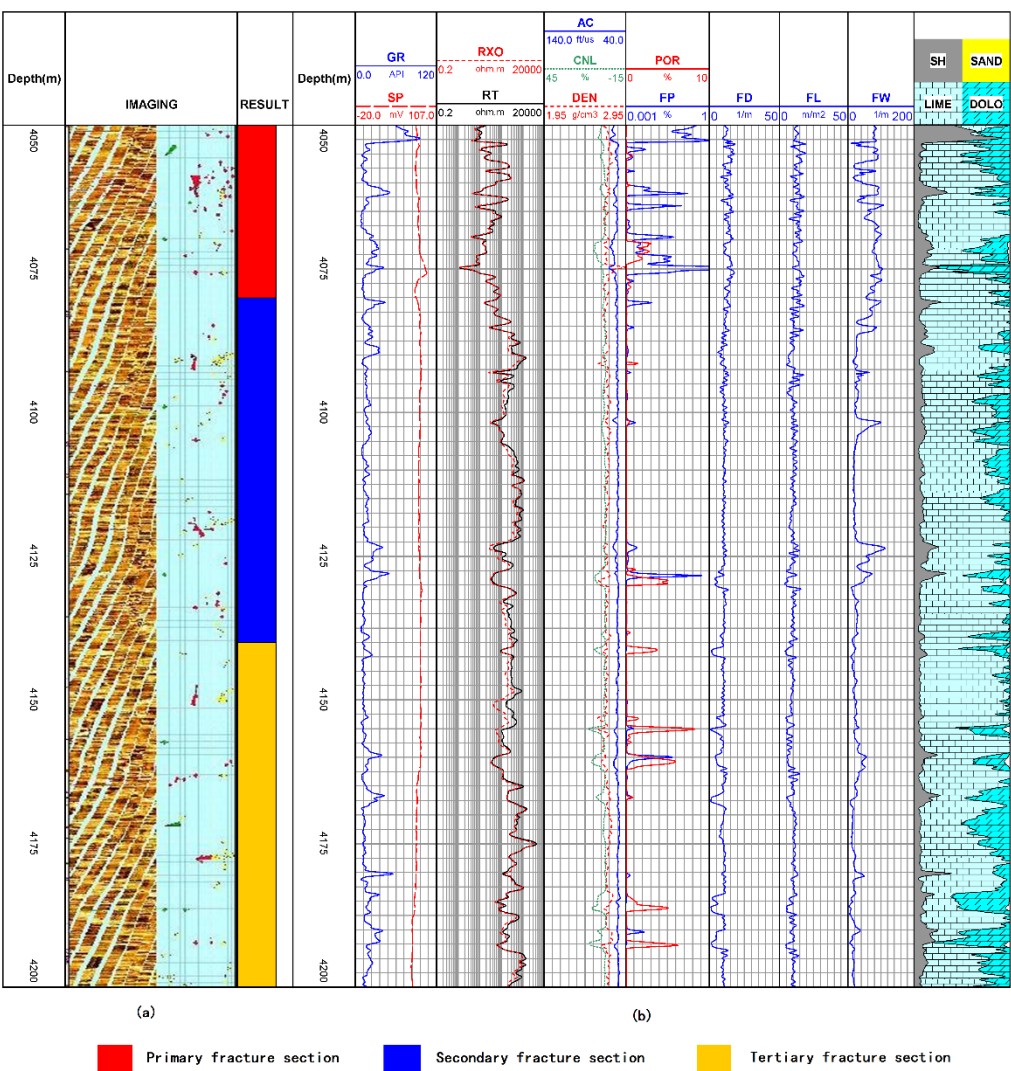

**Figure 6.** Comparison of fracture parameters between imaging logging fracture classification and conventional logging predictions in the Laopu Nan 1 well. (**a**) Natural Gramma, Natural potential and Imaging; (**b**) Conventional log.

## 6. Discussion

### 6.1. Results of Model Training

At present, the methods used to calculate the fracture parameters from logging curves primarily include depth and shallow laterolog calculations and machine learning. The results of the typical algorithms employed in these two methods are compared with those of the MLP algorithm proposed in this paper.

Conventional logs (GR, SP, CAL, RLLD, RLLS, DEN, AC, CNL) are difficult, or at least inaccurate when calculating the fracture density and length. However, the fracture width and porosity can be calculated using conventional logs [10]. The formula for calculating fracture porosity and estimating fracture width from dual laterolog depth and shallow lateral data is as follows:

The PoroDist sonoelectric imaging pore distribution analysis software was used in this study (www.bjgeotech.com, accessed on 1 July 2022).

If *RLLD > RLLS*,

$$\text{FPOR}_{CAL} = \sqrt[m]{\left(\frac{1}{RLLS} - \frac{1}{RLLD}\right) \times R_{mf}/(m-1)} \tag{10}$$

If $RLLD \leq RLLS$,

$$\text{FPOR}_{CAL} = \sqrt[m]{\frac{\left(\frac{1}{RLLD} - \frac{1}{RLLS}\right) \times R_{mf} \times R_w}{R_{mf} - R_w}} \tag{11}$$

$$\text{FW}_{CAL} = \left(\frac{1}{RLLD} - \frac{1}{RLLS}\right) / \frac{1}{R_{mf}} \times 4 \times 10^{-4} \tag{12}$$

where $R_{mf}$ is the resistivity of the mud filtrate, $R_w$ is the resistivity of the formation water, and m is the fracture porosity index, which is 1.1 in this study area.

Figure 7 shows the comparison between the fracture width and the porosity calculated using the depth and shallow lateral data from dual laterolog and the results of the MLP model proposed in this paper. The fracture porosity (FPOR_CAL) and fracture width (FW_CAL) calculated by the MLP model are small, and the fracture width (FW_PRE) and fracture porosity (FPOR_PRE) predicted by the MLP model are in good agreement with the measured values (FPOR, FW). Therefore, dual laterolog depth and shallow laterolog data are not suitable for fracture parameter calculation in the study area.

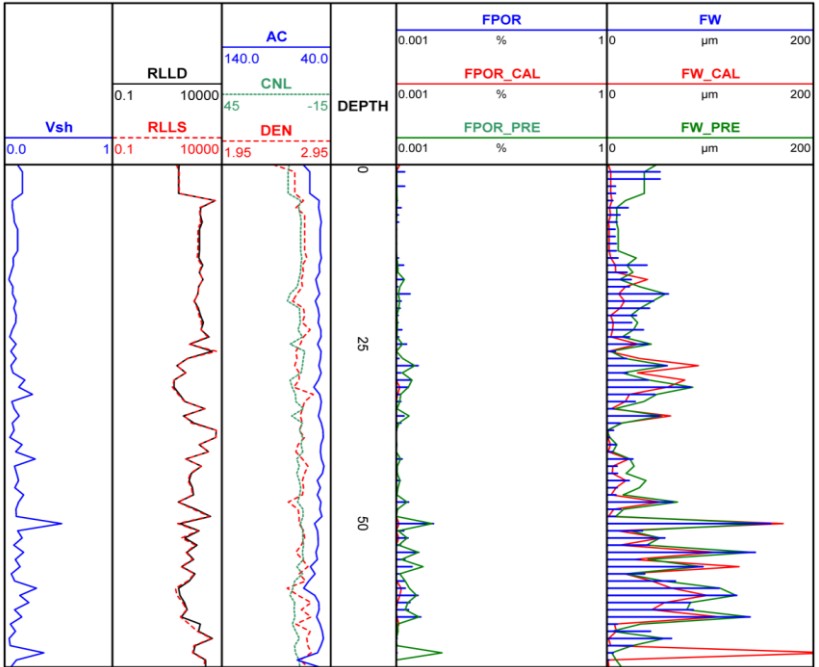

**Figure 7.** Comparison of the fracture width and fracture porosity calculated by dual laterolog and the proposed MLP method.

Machine learning theory and methods have been widely used to solve complex problems in engineering applications and scientific fields. The employed methods primarily include the Random Forest Algorithm (RF), Linear Regression (LR), K-nearest Neighbour (KNN), Support Vector Machines (SVM), Boosting (AdaBoost, Gradient Boosting), and Bagging. These algorithms can be used for classification and regression. As the essence of classification and regression is the same, the classification model discretizes the regression model's output, and regression results in an approximate prediction of the true value.

In this study, Python and machine-learning algorithms from the scikit-learn library (http://scikit-learn.org, accessed on 1 July 2022) were used to model and predict the reservoir fracture parameters, and their results were compared with the MLP model. The accuracy of eight machine learning algorithms in terms of fracture parameter prediction was tested using a validation data set. Figure 8 shows the results of the evaluation, and the MLP model (red column) demonstrated good performance and robustness when predict-

ing all four fracture parameters. In conclusion, compared with other popular models, the established MLP algorithm has higher accuracy and more significant advantages in terms of fracture parameter evaluation.

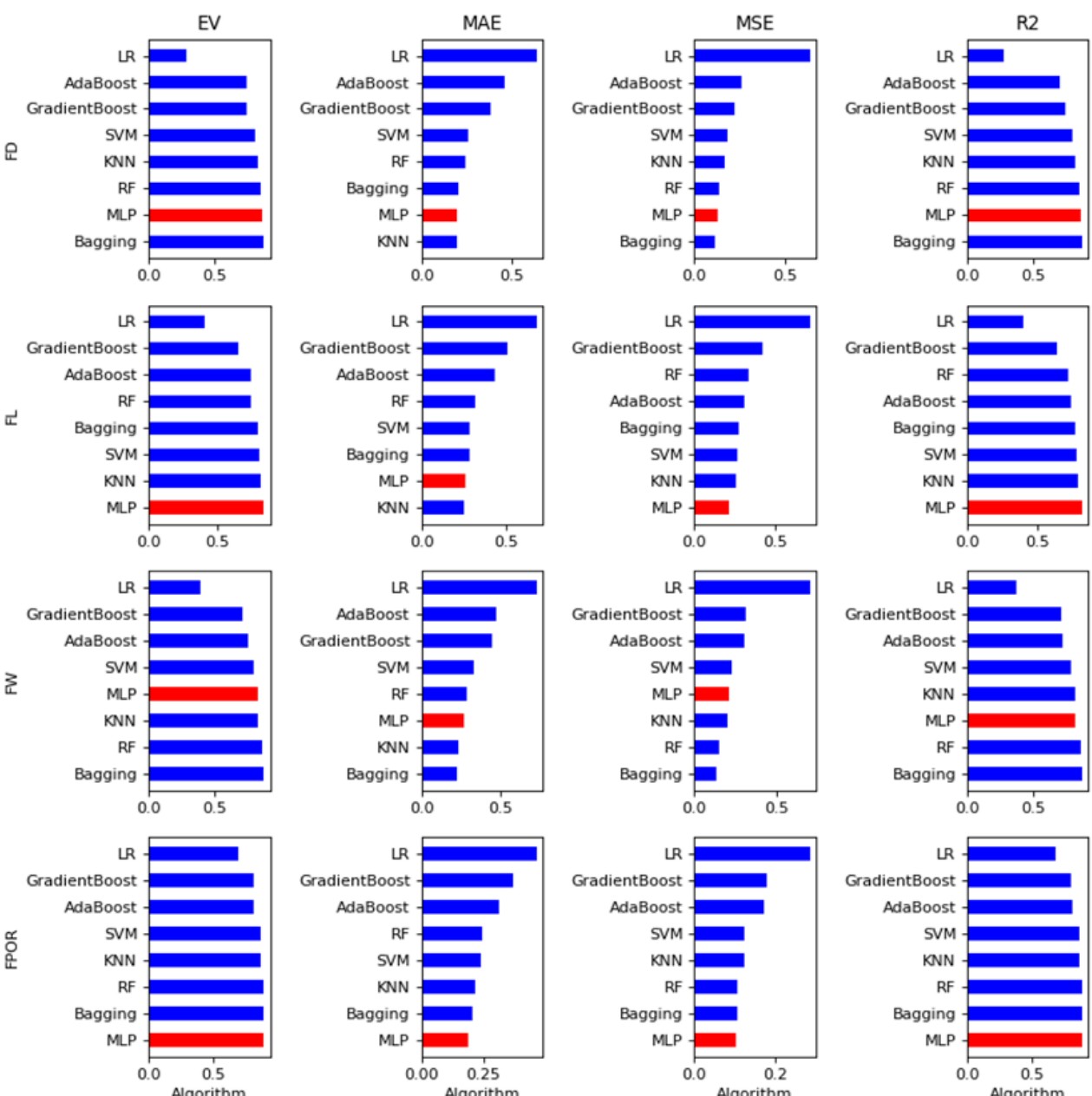

**Figure 8.** Performance evaluation of various machine-learning methods for fracture parameter prediction.

### 6.2. Evaluation of Reservoir Fracture Development

The fracture parameters for nine drilled wells were predicted using trained optimal networks (Table 3). The well test results for B1 show a daily oil yield of 664 tons. Although the reservoir porosity and fracture porosity are small, averaging 1.53% and 0.04%, respectively, the fracture width, length and density of the well are the largest of the nine wells. Well A2's test results show a daily output of 61 tons of water and a small amount of oil. Not only are the related reservoir porosity and fracture porosity smaller, with an average

of 1.34% and 0.12%, respectively, but the fracture width and density are also lower. The influence of multiple fracture parameters on hydrocarbon accumulation is defined as FP:

$$FP = FD \times FW \times FL \tag{13}$$

**Table 3.** Average values of fracture porosity and fracture parameters in the target layer.

| Well Name | STRATUM | OIL | Water | POR | FPOR | FW | FL | FD | Oil Test Conclusion |
|-----------|---------|-----|-------|-----|------|-----|-----|-----|----------------------|
| A1 | Ordovician | 0 | 0 | 2.42 | 0.27 | 16.02 | 7.99 | 4.5 | No display |
| A2 | Ordovician | 0 | 61 | 1.34 | 0.12 | 15.25 | 10.21 | 6.04 | Low production flow |
| A3 | Ordovician | 57 | 326 | 4.21 | 0.57 | 26.68 | 9.34 | 5.82 | Industrial oil flow |
| A4 | Ordovician | 0 | 6.88 | 2.22 | 0.01 | 23.81 | 7.43 | 7.11 | Low production flow |
| A5 | Cambrian | 1.98 | 15 | 4.21 | 0.42 | 30.04 | 6.85 | 6.2 | Industrial oil flow |
| B1 | Ordovician | 664 | 13 | 1.53 | 0.04 | 10.31 | 10.31 | 10.21 | Industrial oil flow |
| B2 | Ordovician | 80 | 0 | 2.03 | 0.06 | 49.69 | 9.6 | 8.57 | Industrial oil flow |
| B3 | Ordovician | 0.66 | 102 | 1.53 | 0.25 | 25.85 | 9.35 | 6.11 | Industrial oil flow |
| B4 | Ordovician | 64 | 0 | 8.18 | 0.05 | 58.42 | 6.07 | 5.83 | Industrial oil flow |

Figure 9 shows the cross-plot of the FPOR and FP. Among them, the black dots represent industrial oil flow wells, the blue dots are low-yield oil flow wells, and the red dots are well that are not shown. Figure 4 shows that the larger the FP is, the better the oil and gas display is; additionally, the larger the FPOR, the better the gas display. In addition to fracture porosity, fracture width, length and density are also important parameters when determining the availability of reservoir space, as they reflect the complexity of fluid accumulation in carbonate formations. The application of the MLP fracture parameter prediction model in the study area is well verified by the oil test data, and the evaluation results are reliable. In addition, new well predictions can be made based on this chart.

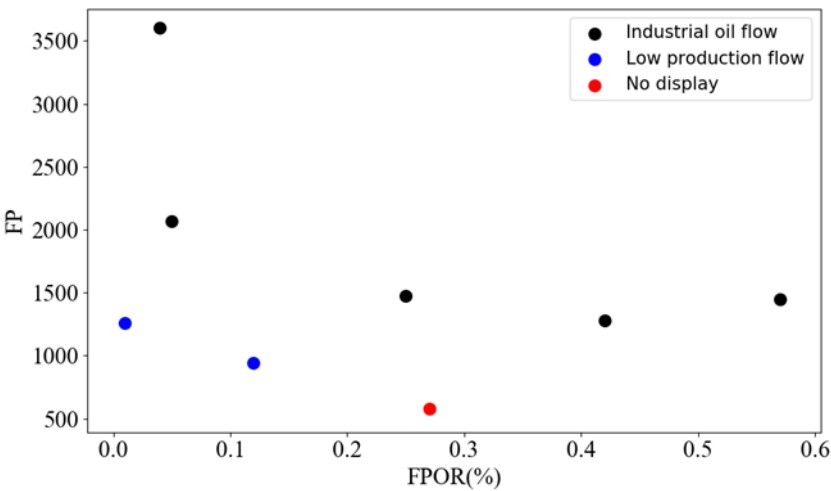

**Figure 9.** Cross-plot of the FPOR and FP.

The study area consists primarily of limestone, dolomitic limestone, dolomitic limestone, calcareous dolomite, dolomitic limestone, dolomite, marl, and other rock types. The results show that the shale content is higher at the fracture height. Previous studies have shown that fractures are highly developed in the fault core and damage zone, while fracture density decreases in the direction opposite the main fault zone. Therefore, there may be a relationship between a higher-than-average shale content and fracture density in carbonate rocks. Previous studies have concluded that the reasons for this phenomenon are a shear drag, grinding, pressure dissolution, and dissolution of marl along the fault section during fault activity, as well as the addition of supergenetic karst infiltration clay. There-

fore, it is very important to study the fracture development degree of rocks with different mineral components to further define key exploration targets.

According to the MLP fracture parameter prediction model established in this paper, the development degree of fractures in different rock types can be predicted by using the response values of acoustic time differences and neutron density logging in limestone and dolomite. The specific steps needed to implement this process are as follows:

(1) The average resistivity of the strata in the study area is 500Ω.m. For pure limestone and pure white dolomite, the log value of the acoustic time difference is 47.5 us/ft and 43.5 us/ft, respectively. The density logging values were 2.71 g/cm$^3$ and 2.87 g/cm$^3$, respectively;

(2) The acoustic time difference and density logging values of different shale content (Vsh) and dolomite content are simulated according to the physical model of rock volume and its logging response equation;

(3) The MLP model is used to simulate and predict the fracture parameters;

(4) The fracture development degree is evaluated according to the carbonate component classification table.

The prediction results show that: (1) Dolomitic limestone and calcite dolomite show high fracture porosity, width, density, and length when the local argillaceous content is between 0% and 6% (Figure 10). In other words, high-quality reservoirs develop in dolomitic limestone and calcareous dolomite formations. (2) When the argillaceous content is between 6% and 10% and the dolomite content is greater than 40%, the fracture porosity, density, and length are higher, but the fracture width is lower. This observation shows that in the case of high argillaceous content, a reservoir is developed in terms of dolomitic limestone, calcareous dolomite, grey dolomite, and dolomite formations. Due to the filling of fractures, the formation is a sub-level high-quality reservoir (Figure 11). (3) The shale-free zone shows low fracture porosity, density, and length but high fracture width, indicating that fractured reservoirs do not easily develop (Figure 12).

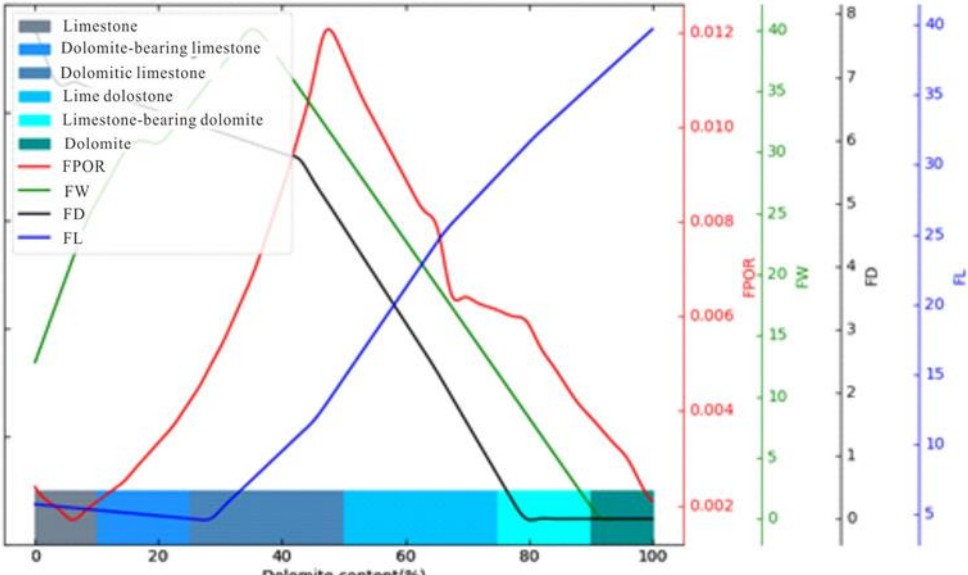

**Figure 10.** Development of fracture parameters with different dolomite contents under the condition of low argillaceous content.

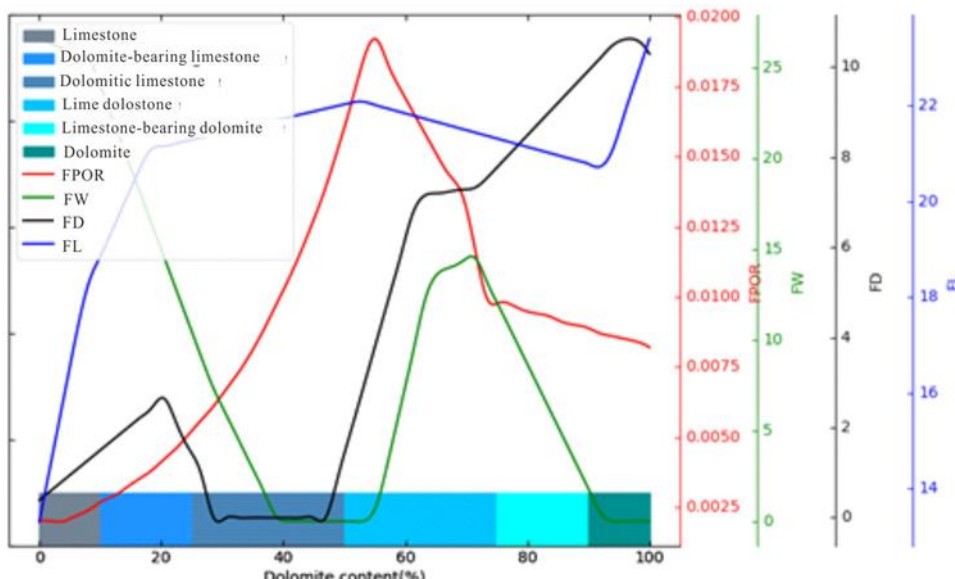

**Figure 11.** Development of fracture parameters with different dolomite contents under the condition of relatively high argillaceous content.

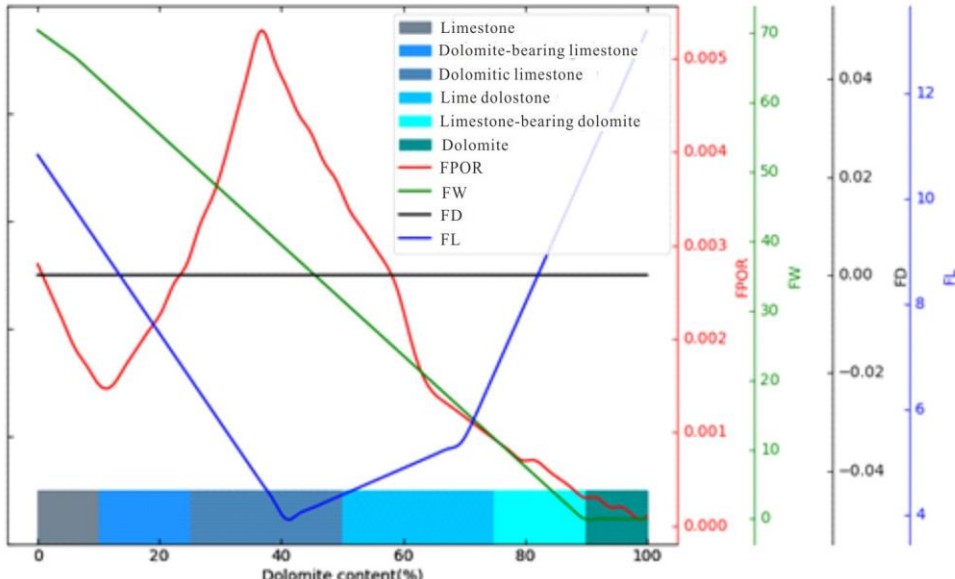

**Figure 12.** Development of fracture parameters with different dolomite contents in the absence of argillaceous content.

Based on the above research results, it can be predicted that the upper Majiagou Formation, Lower Majiagou Formation, Fengshan Formation, and Gushan Formation will develop high-quality reservoirs, which will be key exploration horizons in the next step (Table 4). In addition, the Changshan Formation, Zhangxia Formation, Lu Tou Formation, and Fujunshan Formation developed high-quality secondary reservoirs.

**Table 4.** Prediction of fracture development degree based on the MLP model of mineral composition in geological outcrops.

| Formation | Calcite (%) | Dolomite (%) | Argillaceous (%) | Others (%) | Prediction of Fracture Development Degree | | | | Reservoir Classification |
|---|---|---|---|---|---|---|---|---|---|
| | | | | | FPOR | FD | FW | FL | |
| Fujunshan | 0~95% | 0~98% | 0~34% | A little pyrite | Good | Good | Poor | Good | II |
| Mantou | 0~2% | 61~93% | 4~34% | Pyrite (0~7%) | Good | Good | | Good | II |
| Maozhaung | 98% | | 0.5% | Pyrite (0~5%) | | | | | |
| Xuzhaung | | 1~6% | 78~83% | | | | | | |
| Zhangxia | 83~95% | 0~12% | | Glauconite, Pyrite | Poor | Good | Good | Poor | II |
| Gushan | 25~95% | 3~30% | 2~38% | | Good | Good | Good | Good | I |
| Changshan | 87% | 10% | 3% | A little pyrite | Poor | Good | Good | Poor | II |
| Fengshan | 31~86% | 12~65% | 2~3% | A little pyrite | Good | Good | Good | Good | I |
| Liangjiashan | 72% | 28% | | | Poor | Poor | Good | Poor | II |
| Xiamajiagou | 12~99% | 25~85% | 1~4% | Pyrite (<1%) | Good | Good | Good | Good | I |
| Shangmajiagou | 55~94% | 1.5~43.5% | 0.3~1% | Pyrite (0.5~1%) | Good | Good | Good | Good | I |

## 7. Conclusions

Artificial intelligence and machine learning have become integrated into various research fields, and they gradually assumed a key role in practical applications. Therefore, this paper adopted the MLP algorithm to predict fracture parameters and established a fracture parameter evaluation model that employs well-logging data extracted during drilling. The results show that this algorithm's accuracy can reach 82%. Compared with other conventional fracture parameter calculation or prediction methods, the MLP algorithm has a good application effect in fractured reservoir evaluation. The optimal network model was used to simulate the control effect of different argillaceous contents and dolomite contents on the fracture parameters, and it was found that argillaceous content has a considerable influence on fracture development, not purely dolomite. Future research will include increasing the data volume, improving the accuracy of the evaluation models, and exploring fracture permeability prediction methods. The MLP model employed in this paper is suitable for the prediction of fracture parameters in the target region, and the specific modelling steps and analysis method can provide a reference for similar research. In addition, the goodness of fit scores were greater than 0.97 and 0.82.

(1) Compared with traditional research methods, this paper adopted data-driven methods as the core, which were supplemented by the MLP machine-learning algorithm, to achieve the prediction of intra-type favourable reservoirs based on the fracture origin against the backdrop of the ultra-low physical properties of carbonate rocks.

(2) The results show that this algorithm's accuracy can reach 82%. Compared with other conventional fracture parameter calculation or prediction methods, the MLP algorithm has a good application effect in fractured reservoir evaluation. The optimal network model was used to simulate the control effect of different argillaceous content sand dolomite contents on the fracture parameters, and it was found that argillaceous content has a considerable influence on fracture development, not purely dolomite.

(3) The research method in this paper integrated geological principles and computer and logging technology to realize interdisciplinary advantages, broaden the research methods used in petroleum geological exploration and development, and provide a reference for other similar research. In addition, the application of this research method in other oil fields needs to be combined with the basic data of the corresponding oil fields and make corresponding adjustments in practical applications.

**Author Contributions:** Conceptualization, J.P.; methodology, J.P. and Y.Z.; software, J.P.; validation, J.P.; investigation, J.P.; writing—review and editing, J.P.; project administration, J.P.; funding acquisition, Y.Z.; All authors have read and agreed to the published version of the manuscript.

**Funding:** This research was funded by the National Natural Science Foundation of China (grant number 42172150).

**Data Availability Statement:** Not applicable.

**Conflicts of Interest:** The authors declare no conflict of interest.

## Nomenclature

The format in the attached table includes the full name (abbreviation, none or unit).

| | |
|---|---|
| MLP | Multi-Layer Perceptrons |
| FPOR, % | Fracture Porosity |
| FMI | Fullbore Formation Microresistivity Scanning Imaging Logging |
| DEN, g/cm$^3$ | Neutron Density |
| RF | Random Forest Algorithm |
| RILD/RLLD, $\Omega\bullet$m | Deep/shallow investigation induction log |
| FD, 1/m | Fracture Density |
| FO | Fracture Orientation |
| GRs, API | Gamma Rays |
| EVS | Explained Variance Score |
| LR | Linear Regression |
| RLLS | Deep investigate double lateral resistivity log |
| FL, m/m$^2$ | Fracture Length |
| FDA | Fracture Dip Angle |
| ACs, us/m | Acoustic Curves |
| AC | Acoustic Logging |
| MAE | Mean Absolute Error |
| KNN | K-nearest Neighbour |
| SP | Spontaneous Potential |
| FW, $\mu$m | Fracture Width |
| FF | Fracture Filling |
| CNL | Neutron Porosity Logging |
| MSE | Mean Squared Error |
| SVM | Support Vector Machines |
| CAL, ln/cm | Caliper Log |

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
