# Peer review of "Prediction of Reservoir Fracture Parameters Based on the Multi-Layer Perceptron Machine-Learning Method: A Case Study of Ordovician and Cambrian Carbonate Rocks in Nanpu Sag, Bohai Bay Basin, China"

_processes, doi:10.3390/pr10112445_

Round 1

Reviewer 1 Report

My comments: 1. For MLP models the authors use data from only 3 wells and 99 points of logging data. Is this enough to create a high quality MLP model? As far as I know, for the application of MLP algorithms, it is better to use more data (for example - 1000 and more data samples), and for a small amount of data, it is better to use simpler algorithms. 2. Figure 1 does not show all the wells used in this study. I recommend showing all wells which authors use for this study on this figure. 3. I recommend that authors indicate the units of measurement for parameters in all tables of this article.

Author Response

Comments of reviewer:

  1. For MLP models the authors use data from only 3 wells and 99 points of logging data. Is this enough to create a high quality MLP model? As far as I know, for the application of MLP algorithms, it is better to use more data (for example - 1000 and more data samples), and for a small amount of data, it is better to use simpler algorithms.

Reply: This study area is still in the early stage of exploration and development, and a large number of follow-up relevant data are being gradually supplemented, which will take some time. Thank you for your constructive suggestions.

  1. Figure 1 does not show all the wells used in this study. I recommend showing all wells which authors use for this study on this figure.

Reply: The map has been supplemented with relevant well location information

  1. I recommend that authors indicate the units of measurement for parameters in all tables of this article.

Reply: It has been modified and the full names of all abbreviations are summarized at the end of the article (see Schedule 1).

Reviewer 2 Report

The review of the manuscript entitled: “Prediction of reservoir fracture parameters based on the multilayer perceptron machine learning method: A case study of Ordovician and Cambrian carbonate rocks in Nanpu Sag, Bohai Bay Basin, China”. In this manuscript, the authors used a multi-layer perceptron algorithm to predict fracture parameters and established a fracture parameter evaluation model that employs well logging data extracted during drilling. The work is very interesting and has a scientific style. There are some general comments and questions before its publication as follows:

1.      The work is a case study. How this study can be used in other oilfields? It should be mentioned.

2.      Abstract: this section is different from the Introduction section. The main findings with important opinions are acceptable. The authors need to consider these points in the revision stage.

3.      There are many abbreviations. Please add a table for abbreviations to easily follow them.

4.      In Table 5, for the prediction of fracture development degree, instead of good, or poor, it is recommended to evaluate it quantitatively.

5.      In the introduction section, it is recommended to add a sentence about the production control in oilfields such as asphaltene control. For this purpose, the next reference can be used in the revision stage: “The control of asphaltene precipitation in oil wells, Petroleum Science and Technology”, 36(6), 443-449;   

6.      The authors introduced the error in section 4.4. But they did not mention it in the results and discussions. Please consider it in the revision stage.

7.      Please support the obtained results with references.

Author Response

Comments of reviewer:

The review of the manuscript entitled: “Prediction of reservoir fracture parameters based on the multilayer perceptron machine learning method: A case study of Ordovician and Cambrian carbonate rocks in Nanpu Sag, Bohai Bay Basin, China”. In this manuscript, the authors used a multi-layer perceptron algorithm to predict fracture parameters and established a fracture parameter evaluation model that employs well logging data extracted during drilling. The work is very interesting and has a scientific style. There are some general comments and questions before its publication as follows:

  1. The work is a case study. How this study can be used in other oilfields? It should be mentioned.

Reply: The relevant content is supplemented at the end of the article. The application examples of other oil fields are in progress, and the final application results have not been formed yet. In addition, the application of this research method in other oil fields needs to be combined with the basic data of the corresponding oil fields, and make corresponding adjustments in practical application.

  1. Abstract: this section is different from the Introduction section. The main findings with important opinions are acceptable. The authors need to consider these points in the revision stage.

Reply: It has been further supplemented in the abstract

  1. There are many abbreviations. Please add a table for abbreviations to easily follow them. Reply: It has been modified and the full names of all abbreviations are summarized at the end of the article (see Schedule 1).
  2. In Table 5, for the prediction of fracture development degree, instead of good, or poor, it is recommended to evaluate it quantitatively.

Reply: The quantitative evaluation criteria for fracture prediction are being established, and the relevant results will be elaborated in the following papers. This paper focuses on the establishment of forecasting methods. Thank you for your forward-looking suggestions.

  1. In the introduction section, it is recommended to add a sentence about the production control in oilfields such as asphaltene control. For this purpose, the next reference can be used in the revision stage: “The control of asphaltene precipitation in oil wells, Petroleum Science and Technology”, 36(6), 443-449;

Reply: The relevant content has been supplemented in the introduction and references.

  1. The authors introduced the error in section 4.4. But they did not mention it in the results and discussions. Please consider it in the revision stage.

Reply: The relevant content is discussed in the conclusion and first paragraph of 5.1

7. Please support the obtained results with references.

Reply: The relevant results support is forming an internal report, which has not yet been made public. The author will gradually demonstrate the application results of this method in detail in the future. Thank you for your constructive suggestions.

Round 2

Reviewer 2 Report

The work was revised based on the comments. So, in my opinion, the work is ready for publication.